# Enhancer mutations modulate the severity of chemotherapy-induced myelosuppression

Artemy Zhigulev[1], Zandra Norberg[1], Julie Cordier[1], Rapolas Spalinskas[1], Hassan Bassereh[1], Niclas Björn[2], Sailendra Pradhananga[1], Henrik Gréen[2,3,*], Pelin Sahlén[1,*]

Non-small cell lung cancer is often diagnosed at advanced stages, and many patients are still treated with classical chemotherapy. The unselective nature of chemotherapy often results in severe myelosuppression. Previous studies showed that protein-coding mutations could not fully explain the predisposition to myelosuppression. Here, we investigate the possible role of enhancer mutations in myelosuppression susceptibility. We produced transcriptome and promoter-interaction maps (using HiCap) of three blood stem-like cell lines treated with carboplatin or gemcitabine. Taking advantage of publicly available enhancer datasets, we validated HiCap results in silico and in living cells using epigenetic CRISPR technology. We also developed a network approach for interactome analysis and detection of differentially interacting genes. Differential interaction analysis provided additional information on relevant genes and pathways for myelosuppression compared with differential gene expression analysis at the bulk level. Moreover, we showed that enhancers of differentially interacting genes are highly enriched for variants associated with differing levels of myelosuppression. Altogether, our work represents a prominent example of integrative transcriptome and gene regulatory datasets analysis for the functional annotation of noncoding mutations.

## Introduction

Cancer, a leading cause of death, is responsible for nearly one in six deaths worldwide (Ferlay et al, 2021). Lung cancer is the second most frequently diagnosed cancer, causing 1.8 million yearly deaths (Thai et al, 2021). Of those, 84% are diagnosed with non-small cell lung cancer (NSCLC) (Ganti et al, 2021), often at an advanced stage because of a lack of clinical symptoms and effective screening approaches (Gridelli et al, 2015). For many patients with advanced-stage NSCLC, where targeted therapies and immunomodulators are not indicated, the main treatment route remains traditional chemotherapy using the third generation chemotherapy (such as paclitaxel, docetaxel, gemcitabine, vinorelbine or irinotecan) in combination with platinum derivatives (carboplatin, cisplatin) (Baggstrom et al, 2007). The classical drug cocktail for NSCLC, gemcitabine in combination with carboplatin (Sederholm et al, 2005), is also widespread for treating other solid tumors such as bladder, ovarian, and breast cancers (Pittman et al, 2006). Despite their widespread application, these drugs can lead to severe adverse drug reactions (ADRs), often resulting in treatment cessation, treatment failure or even death (Testart-Paillet et al, 2007; Amjad et al, 2023).

Blood cell progenitors, located in the bone marrow, are fast-dividing cells that maintain the turnover of blood cells, such as lymphocytes, erythrocytes, platelets, and neutrophils (Carey, 2003). Their untargeted death leads to myelosuppression—dose-limiting toxicity in carboplatin/gemcitabine treatment (Chatelut et al, 2003). Notably, among platinum derivatives, it is carboplatin which is mainly associated with a higher risk of neurotoxicity and myelosuppression, whereas ADRs of cisplatin include a higher rate of nausea, vomiting, nephrotoxicity, and ototoxicity (Santana-Davila et al, 2014). So far, only neutropenia can be partially modulated in some cases by G-CSF (granulocyte colony-stimulating factor) (Pastor et al, 2013). As a result, around 55% of the patients receiving chemotherapy stopped it because of severe or life-threatening myelosuppression levels, even if the drug is effective for treating the tumor (Zatloukal et al, 2003; Sederholm et al, 2005; Rudd et al, 2005; Grønberg et al, 2009; Imamura et al, 2011). Therefore, it is imperative to devise tools to stratify patients with respect to their predisposition to experiencing ADRs, especially chemotherapy-induced myelosuppression. Genetic factors heavily modulate drug response phenotype, so patient genotype information could be used for therapy management and optimization (Trendowski et al, 2019; Mulford et al, 2021). A recent study by Swen et al (2023) validated this hypothesis and reported the use of a pharmacogenetics panel containing 12 genes. The use of the panel significantly reduced the number of patients with ADRs and was feasible across European healthcare system organizations (Swen et al, 2023).

[1]Royal Institute of Technology - KTH, School of Chemistry, Biotechnology and Health, Science for Life Laboratory, Stockholm, Sweden  [2]Division of Clinical Chemistry and Pharmacology, Department of Biomedical and Clinical Sciences, Faculty of Medicine and Health Sciences, Linköping University, Linköping, Sweden  [3]Department of Forensic Genetics and Forensic Toxicology, National Board of Forensic Medicine, Linköping, Sweden

Correspondence: pelin.akan@scilifelab.se; artemy.zhigulev@scilifelab.se
*Henrik Gréen and Pelin Sahlén are senior authors

The potential of using germline genetic markers for toxicity response prediction in the context of chemotherapy-induced myelosuppression was already reported in a set of our previous studies. After performing exome-sequencing of 215 NSCLC patients under carboplatin/gemcitabine treatment (Björn et al, 2020b; Svedberg et al, 2020) and whole-genome sequencing of 96 patients displaying the most extreme toxicity levels (neutropenia, leukopenia, and thrombocytopenia) (Björn et al, 2020a), very few coding variants were found to be associated with toxicity response. Almost all toxicity-associated variants were located in noncoding regions, suggesting a possible role for distal *cis*-regulatory elements in toxicity response regulation.

Enhancers are best-studied distal *cis*-regulatory elements containing clusters of binding sites for transcription factors (Banerji et al, 1981; Kleinjan & van Heyningen, 2005). Their disruption was shown to be a disease-driving or contributing mechanism via modulation of the expression of target genes, establishing a new disease group called enhanceropathies (Claringbould & Zaugg, 2021). The chromatin immunoprecipitation (ChIP) method can effectively locate active enhancers using antibodies against the relevant epigenomic marks such as H3K27Ac and H3K4me1 (Liang et al, 2004; Heintzman et al, 2009; Rada-Iglesias et al, 2011). However, their location provides limited insight into the functionality of enhancers (Blackwood & Kadonaga, 1998). Historically, enhancers were linked to their nearest gene. Still, various novel experimental assays showed that only one-third of the enhancers regulate their closest gene (Dostie et al, 2006; Åkerborg et al, 2019), necessitating methods to locate their target genes and their location within the same experiment.

Chromosome conformation capture for high-throughput sequencing (Hi-C) is the chief method to map the spatial conformation of genomes that mediates the promoter–enhancer contacts through looping (Lieberman-Aiden et al, 2009). However, its resolution does not allow for mapping interactions between individual promoter and enhancer elements because of the genome's vast number of structural interactions and stochastic proximities (Forcato et al, 2017). The HiCap technology solves this problem by introducing a sequence capture step on the Hi-C material, enriching for interactions that involve specifically targeted regions (Sahlén et al, 2015). HiCap provides close to single-enhancer resolution (circa 860 bp), outperforming similar methods (Ma et al, 2015; Mifsud et al, 2015; Schoenfelder et al, 2015). This allows fine-mapping of sequence variants and target gene discovery in the same experiment (Åkerborg et al, 2019; Pradhananga et al, 2020; Sahlén et al, 2021; Zhigulev et al, 2022a). Because most variants associated with drug-induced myelosuppression were found within regulatory elements (Björn et al, 2020a), we reasoned that HiCap targeting all the promoters and selected mutations could help prioritize variants for clinical use and provide insights into the mechanisms by which they contribute to toxicity.

## Results

### HiCap depicts interactome dynamics of blood stem-like cells upon drug exposure

We processed 0.8 and 1.1 billion reads mapped on the transcriptome and interactome of 18 samples corresponding to normal and drug-treated counterparts of the three cell lines: CMK, MOLM-1, and K-562, in two replicates. These cell lines are a good and convenient surrogate model readily available in many research laboratories (Skopek et al, 2023). We first investigated gene expression changes upon drug exposure. The technical replicates of the experiments correlated well in all cases (coefficient of correlation >0.93 across all conditions) (Fig S1A–F). However, RNA profiles of samples were not always able to separate treated cells from their normal counterparts (Fig S2). Nevertheless, we detected differentially expressed (DE) genes (FDR < 0.1), absolute log-fold change (abs[logFC] > 1.2) belonging to relevant processes for drug exposure such as regulation of hemopoiesis (GO:1903706, FDR = 1 × 10$^{-8}$) and response to toxic substances (GO:0009636, FDR = 5 × 10$^{-4}$) (Fig S3A and B, Table S1), confirming the cells' response to the treatment.

Next, we analyzed HiCap datasets to find genomic interactions of targeted promoters and selected variants (see the Materials and Methods section). We detected 114,365 distal regions (i.e., untargeted regions that are located distal to the promoters) interacting with 7,254 promoters (supporting pairs >4 and Bonferroni-adjusted *P*-value < 0.01) across all cell lines and treatments (Table S2). Promoter-interacting regions (PIRs) covered around 2.8% (84.4 Mb) of the genome. We also detected 18,024 interactions between promoters and 19,204 interactions between targeted variants and distal regions. Because K-562 is a tier 1 ENCODE cell line, a vast array of public ATAC-seq and ChIP-seq datasets is available (Table S3). We used these datasets to assess the enhancer potential of the PIRs. In K-562 cells, 51.2% (22,719/44,366) of the PIRs overlapped with at least one enhancer mark (6.52-fold enrichment) (Fig 1A). We found that 41% (18,190/44,366) of the PIRs overlapped only a TF-binding site, highlighting the importance of TF datasets during the evaluation of promoter–enhancer interactions. All super-enhancers (SEs) annotated in K-562 cells (Jiang et al, 2019) also overlapped with at least one PIR.

We further tested the potential of detected PIRs to modulate the activity of targeted genes using an inducible dual-effector epigenetic interference system enCRISPRi-LK (Li et al, 2020). We successfully reduced the gene expression in two out of three cases by repressing their enhancers (Fig 1B and C). The first gene, *TGFBR1* (Transforming Growth Factor *β* Receptor 1), is a part of the transforming growth factor *β* signaling pathway known for supporting the maintenance of the self-renewal capacity of hematopoietic stem cells (Blank & Karlsson, 2015). It is regulated by an enhancer located 272 kb away in the chr9:101,595,333–101,595,751 region (Fig S4A). The second gene, *ANAPC4* (Anaphase Promoting Complex Subunit 4), is part of a highly conserved protein complex that controls the amounts of the cyclins and other cell cycle regulators to ensure proper cell cycle transitions (Wäsch et al, 2010). Enhancer of *ANAPC4* is located 326 kb away from it in the chr4:25,051,758–25,052,804 region (Fig S4B). None of these regions interact with *NFE2* (Nuclear Factor Erythroid 2) promoter used as a negative control.

We then investigated different aspects of interaction data by looking at interaction statistics, functional enrichment of interacting regions, and expression profiles of interacting promoters. The average interaction distance across all the samples is 156 kb (Fig 2A). Our interactome maps showed close to single-enhancer resolution (average PIR length of 860 bp) and therefore facilitated the individual discovery of shared, and cell type-specific interactions (Fig 2B).

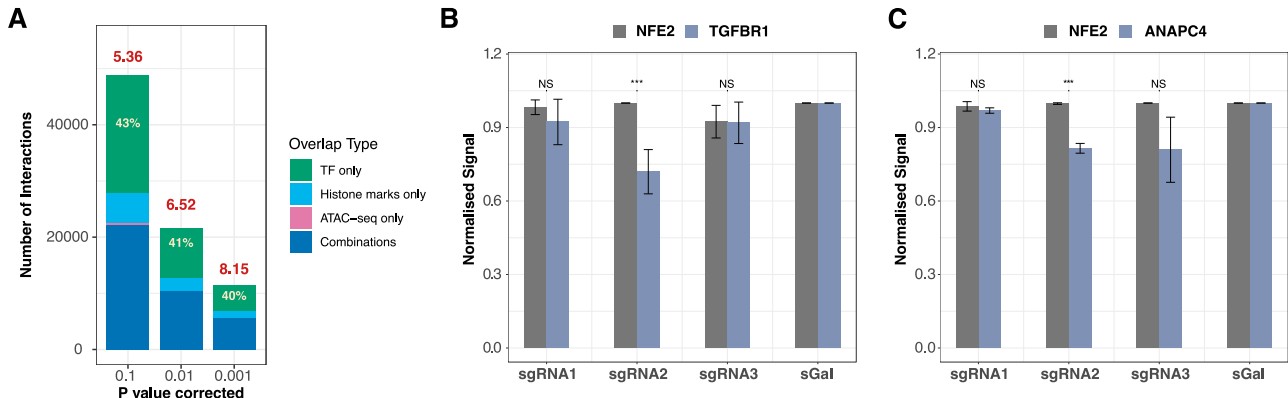

**Figure 1. Validation of K-562 PIRs detected by HiCap.**
**(A)** The enhancer element enrichment profile of PIRs at varying Bonferroni-corrected *P*-value thresholds. The numbers in red denote the fold enrichment for all enhancer elements using the BEDTools fisher tool. The percentages in white show the ratio of PIRs that only overlaps with TF binding sites. **(B, C)** enCRISPRi-LK validation of (B) chr9:101,595,333–101,595,751 region interacting with the *TGFBR1* promoter and (C) chr4:25,051,758–25,052,804 region interacting with the *ANAPC4* promoter. In both cases, different sgRNA2s show a significant effect. sGal represents non-targeting sgRNA. Data are presented as mean ± stdev. ***P ≤ 0.05 (*t* test).

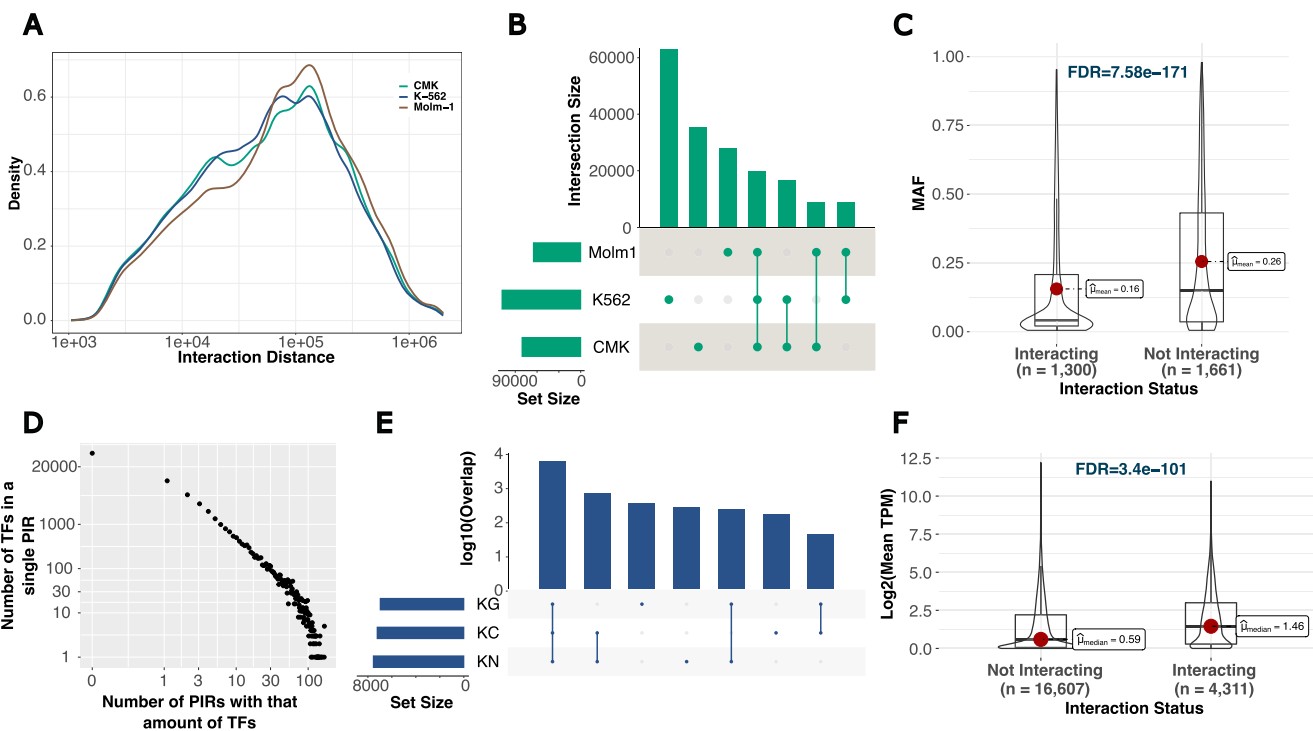

**Figure 2. Interactome organization across different cell lines and treatments.**
**(A)** Absolute interaction distance distribution of all interactions. **(B)** The overlap of interactions across cell types. **(C)** Dependence of targeted SNVs with different MAFs on their interaction status. **(D)** The distribution of the number of TFs found in each PIR. **(E)** The overlap statistics of interacting genes in K-562 cells (K) between treatments: carboplatin (C), gemcitabine (G), or no drug (N). **(F)** Dependence of gene expression on the interaction status of genes.

Interactome profiles were more effective in separating untreated cells from their treated counterparts compared with that of the transcriptome data (Fig S2). Interestingly, 1,302 (43.9%) of targeted variants showed at least one interaction in one cell type, and interacting variants had significantly lower minor allele frequencies compared with those targeted variants without an interaction (Fig 2C, minor allele frequencies = 0.16 versus 0.26, FDR = 7.58 × 10^{-171}).

The ChIP-seq profiles of 218 TFs from K-562 cells (Table S3) were used to investigate the TF distribution across PIRs. The number of TF-binding sites per PIR followed an inverse power–law distribution; that is, most PIRs contained few TFs, and few PIRs contained many TFs (Fig 2D). Using the k-means clustering algorithm, we observed four main TF clusters that co-occur, whereas generally the binding patterns of most TFs were not correlated (Fig S5) suggesting a diversity of enhancers based on their TF-binding patterns. One of

the clusters contained 49 TFs that are highly enriched for SWI/SNF superfamily (FDR = $6.3 \times 10^{-16}$), NuRD complex (FDR = $3.8 \times 10^{-13}$), and leukemogenesis (FDR = $5.8 \times 10^{-11}$). That is in line with previous studies showing a high mutation rate of chromatin remodeling complexes in cancer (Bracken et al, 2019). We then separated PIRs connected to either expressed (top 75% quantile, 16,092 interactions) or not-expressed (bottom 25% quantile, 16,241 interactions) genes to see if there are any differential TF binding between the two groups. There were 52 and 24 TFs binding to PIRs connecting to high or low-expressed genes, respectively, and 23 TF were shared (Table S4). Transcriptional repressor DEAF1 (Michelson et al, 1999) was only found in PIRs connected to low-expressed genes. Moreover, 29 TFs were only found in PIRs connected to expressed genes (Fig S6A and B). Among them, members of the histone deacetylase family (HDAC1, HDAC2, EP400, NCOR1, and TBL1XR1) were present, confirming the role of histone deacetylation in gene activation (Jepsen et al, 2000; Bhaskara et al, 2008; LaBonte et al, 2009; Wang et al, 2009; Seto & Yoshida, 2014). Lastly, we looked at the TF-binding profiles of PIRs as a function of their overlap with SEs. Members of activator complexes such as EP300, EP400, TBL1XR1, NR2C2, and SOX6 were only found in PIRs overlapping with SEs (Fig S6C). However, CTCF and RAD21 (subunit of the cohesin complex [Hauf et al, 2001]) were found only in PIRs not overlapping with SEs (Fig S6D), which supports their role in regulating loop stability (Hansen et al, 2017).

Finally, we focused on the targeted genes across all cell lines and treatments. There were 8,251, 6,582, and 8,142 interacting genes in CMK, MOLM-1, and K-562 cells, respectively. We discovered cell type-specific and treatment-specific interacting genes (Figs 2E and S7A and B). In all cells, they were more likely to be expressed (Fig 2F). However, they were not more likely to be differentially expressed, supporting the different dynamics of the transcriptional regulation by enhancers and promoters (Larsson et al, 2019).

## Network analysis of interactome facilitates the discovery of differentially interacting genes

To track the dynamics of promoter–enhancer interactions, we constructed networks using promoters and PIRs as network nodes and connected them with an edge in case of interaction (see the Materials and Methods section). We generated a separate network for each replicate of cell type and state. We then calculated two parameters, namely, Overlap Coefficient (OCE) and Jaccard Similarity Index (JI), to quantify the connectivity differences for each node across different states for each cell type. If the OCE/JI is equal to one, it means that the gene did not change its interactions. In contrast, if the OCE/JI equals zero, then it means that the gene either changed all interactions, gained all new interaction(s) or lost all its interaction(s) upon drug exposure. As an example, *EEFG1* (Eukaryotic Translation Elongation Factor 1 Gamma) is a housekeeping gene involved in the translation mechanisms (Kumabe et al, 1992), and its interaction profile did not change significantly in CMK cells upon carboplatin induction (Fig 3A). Meanwhile, *EYA3* (EYA Transcriptional Coactivator And Phosphatase 3) plays a particular function as a distinguishing mark between apoptotic and repair responses to genotoxic stress (Cook et al, 2009; Krishnan et al, 2009). It entirely changed the interaction profile under the same conditions (Fig 3B). We named the promoter nodes that changed their

connectivity as differentially interacting (DI) genes (see the Materials and Methods section). This approach revealed hundreds of genes with interaction changes upon treatment despite their relatively even steady-state expression levels (Table S5).

In CMK cells, there were 156/45 DE genes and 728/696 DI genes upon carboplatin and gemcitabine treatments, respectively. Generally, in CMK cells, only DI genes showed enrichments for hematological cell count traits. Therefore, we assessed the biological relevance of DI genes by comparing their enrichment for human phenotype terms with that of non-DI genes, that is, genes that did not change their interactions upon treatment. Indeed, DI genes in CMK cells were more enriched for relevant phenotypes (Fig 3C). For example, *ETV6* (ETS Variant Transcription Factor 6) is a transcriptional repressor implicated in dominantly inherited thrombocytopenia (Hock & Shimamura, 2017). Despite no significant changes in steady-state expression levels, it underwent considerable interactome changes upon both carboplatin and gemcitabine exposure (Fig 3D).

In MOLM-1 cells, we detected 11/252 DE genes and 464/625 DI genes upon exposure to carboplatin/gemcitabine, respectively. They were enriched for relevant traits except for DE genes upon exposure to carboplatin (Fig S8A and B). In particular, *MPL* (Proto-Oncogene, Thrombopoietin Receptor) is essential for the proliferation of megakaryocytes and platelet differentiation (Ng et al, 2014). It was not differentially expressed, but it changed its interaction landscape significantly upon exposure to both drugs (Fig S8C).

In K-562 cells, there were 555/44 DE genes and 921/932 DI genes upon treatment with carboplatin/gemcitabine, respectively. DE genes, upon carboplatin treatment, were enriched for multiple processes related to cell cycle regulation and hematological processes, whereas DI genes showed limited enrichments (Fig S9A). Conversely, very few genes were differentially expressed upon gemcitabine treatment, but DI genes were enriched for multiple processes related to lymphocyte and myeloid cell processes (Fig S9B). *ERG* (ETS Transcription Factor ERG), an essential gene for hematopoiesis (Knudsen et al, 2015), changed both its steady-state expression and interaction status in both treatments (Fig S9C). Concordantly, its haploinsufficiency impairs the self-renewal of hematopoietic stem cells under myelotoxic stress (Ng et al, 2011). The summary of DE and DI genes across cell types and treatments is presented in Table 1.

## Genes connected to candidate enhancer variants tend to differentially interact upon drug exposure

In our previous study, we characterized 8,072,672 single-nucleotide variants (SNVs) from 96 NSCLC patients with differing chemotherapy-induced myelosuppression levels. Only a few protein-coding mutations were associated with the toxicity response of the patients (Björn et al, 2020a). We hypothesized that some SNVs might modulate the activity of enhancers involved in myelosuppression or entirely disrupt them. We overlapped SNVs with 95,567 PIRs derived from HiCap experiments on model cell lines used here to test our hypothesis. Half of the PIRs (52% or 49,802 PIRs) contained at least one variant (50,119 SNVs), and 94.2% of those contained only a single variant.

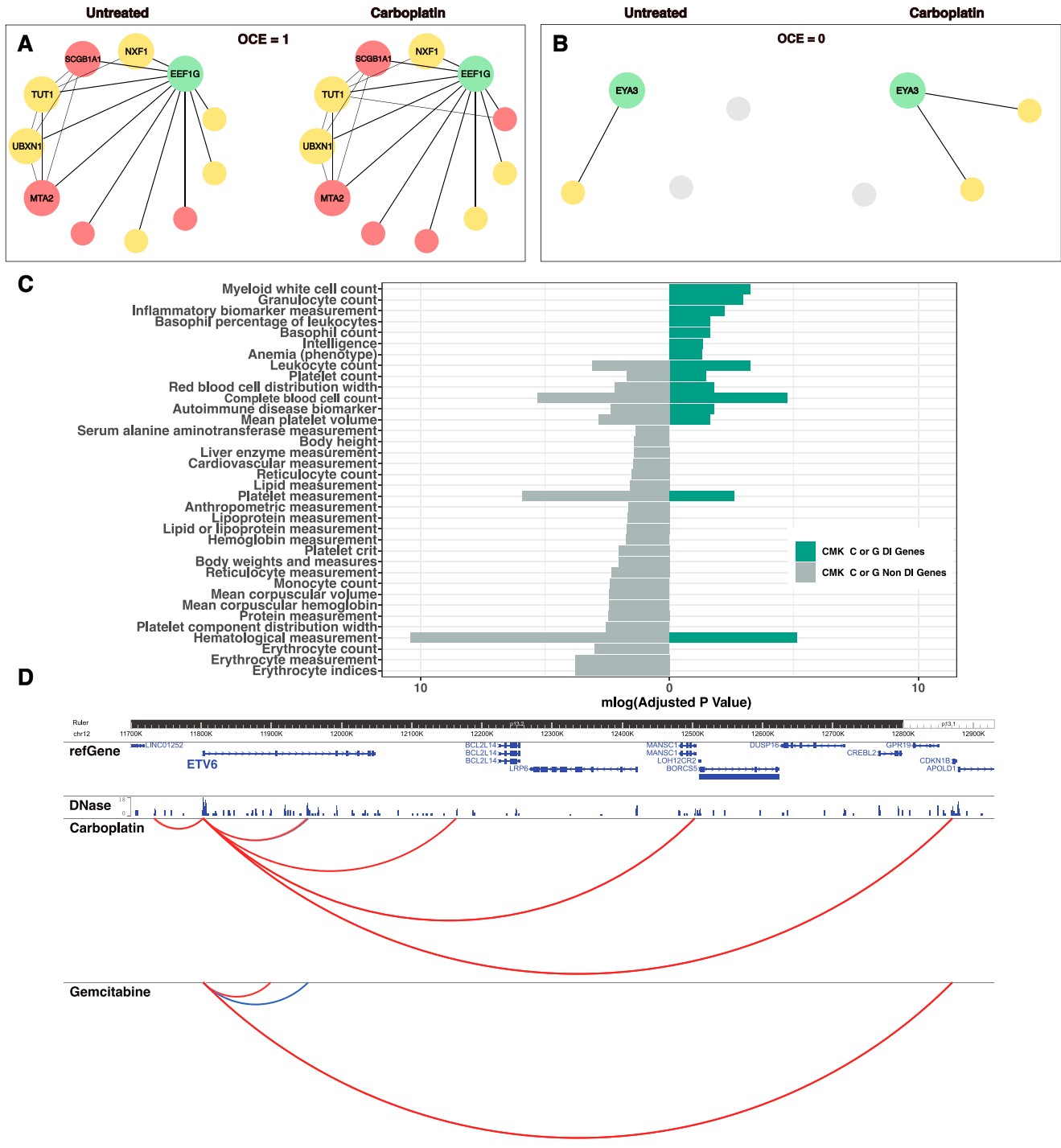

Figure 3. Differentially interacting genes in CMK cells.
**(A, B)** Overlap coefficient measures the similarity between connections of (A) *EEF1G* and (B) *EYA3* in normal versus carboplatin-treated states. Node size reflects its type: gene or enhancer. The color represents the overlap status with DNase hypersensitivity sites: positive (red) or negative (yellow). **(C)** The enriched human phenotype terms in CMK cells treated with either carboplatin or gemcitabine using DI or non-DI genes. **(D)** The interaction profile of *ETV6* upon treatment; static interactions are shown in blue, whereas gained interactions upon treatment are shown in red.

To filter SNVs relevant to higher or lower risk of drug toxicity, we grouped variants based on their allele count (AC) and allele frequency (AF) differences between the two patient cohorts: high toxicity (HT) and low toxicity (LT) (Fig S10). Accordingly, there were 6,583 variants whose AC and AF differed at least by 25% (see the Materials and Methods section). We also required these variants to have a strong effect size (ES) on TF binding affinity. Using the motifbreakR package, we calculated the ES of the selected variants

**Table 1.  Comparison of DE and DI genes across cell lines and treatments.**

| | DE genes | | DI genes | |
|---|---|---|---|---|
| | carboplatin | gemcitabine | carboplatin | gemcitabine |
| CMK | 156 | 45 | 728 | 696 |
| MOLM-1 | 11 | 252 | 464 | 625 |
| K-562 | 555 | 44 | 921 | 932 |

Color reflects enrichment for relevant GO-terms (green–high, orange–medium, red–low).

across a collection of PWMs. Moreover, we filtered only cases where both the interacting gene and TF were expressed (mean [transcripts per million (TPM) across the cell line under all treatments] > 0.2). That prioritized the top 2,720 variants further considered as candidate variants.

Afterward, we asked if promoters connected to PIRs carrying these candidate variants are likelier to change their interaction profile upon drug treatment than those connected to PIRs carrying other (rest) variants. Indeed, promoters interacting with PIRs

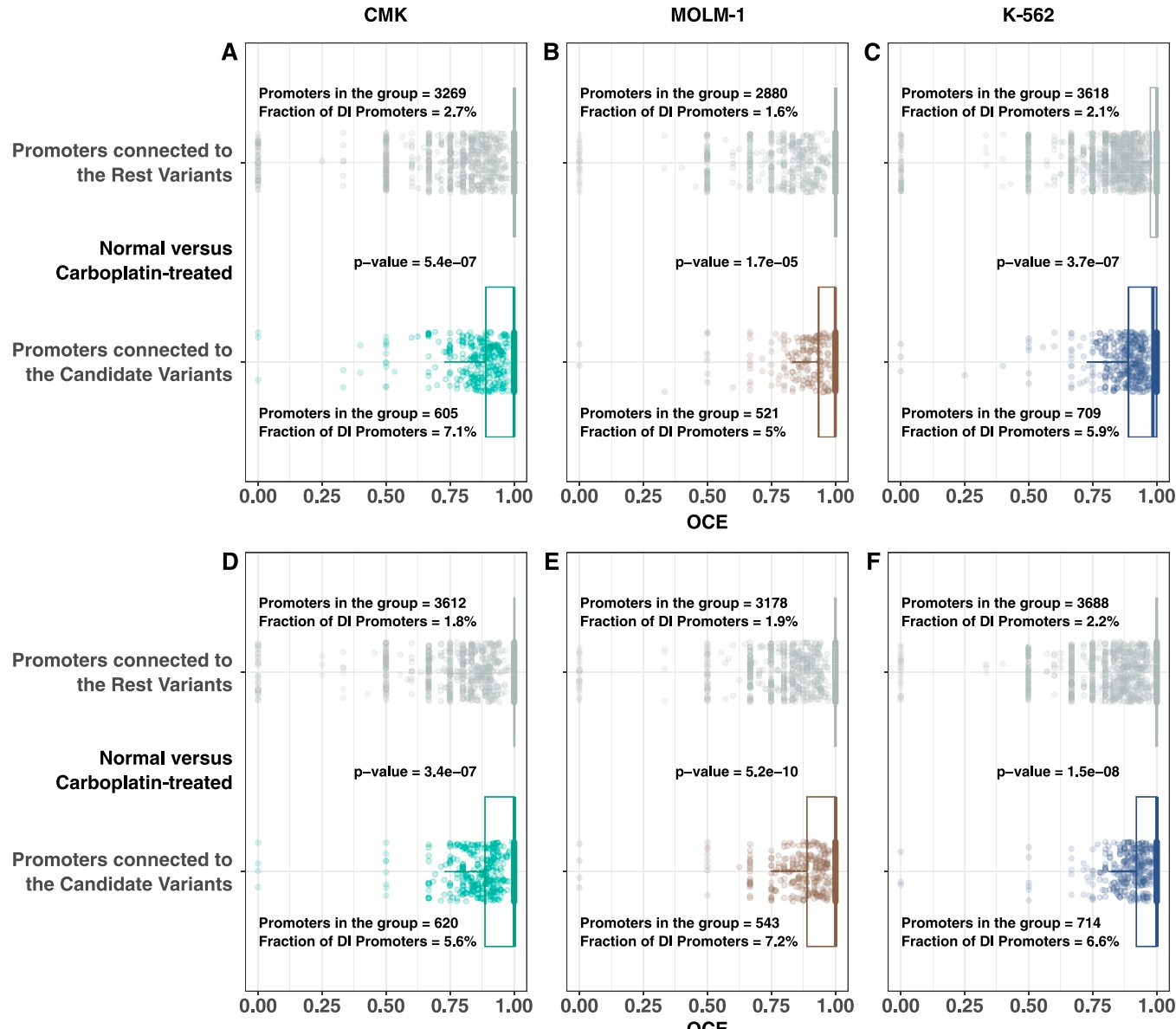

**Figure 4.  Overlap coefficient differences between genes connected to the candidate and rest enhancer mutations.**
**(A, B, C, D, E, F)** Promoters connected to PIRs containing candidate variants are more likely to change their interaction profile upon carboplatin (A, B, C) and gemcitabine (D, E, F) induction than promoters connected to rest variants. Each dot represents a promoter connected to the PIR carrying the patient SNV. Candidate variants were filtered based on several steps described in the main text. The rest corresponds to filtered-out cases. Fisher's exact test was used to calculate *P*-values. Source data are available for this figure.

carrying candidate variants were more likely to change their interactions upon exposure to carboplatin (Fig 4A–C) and gemcitabine (Fig 4D–F). The mean fraction of DI genes carrying variants in PIRs varied from 5.0–7.2% for candidate variants to 1.6–2.7% for the rest cases. The highest P-value between the two groups was $1.7 \times 10^{-5}$ in MOLM-1 cells in normal versus carboplatin-treated cases (Fig 4B), proving that interactions bearing candidate variants are significantly enriched for DI genes. As a result, we report 196 candidate DI genes connected to the candidate variants, which may be involved in the genetic mechanisms of chemotherapy-induced myelosuppression (Table S6).

We further investigated the relevance of the candidate gene set for the myelosuppression. We isolated genes associated (through mutations overlapping genes only) with the counts of lymphocytes, erythrocytes, platelets, and neutrophils from all associations NHGRI-EBI GWAS Catalogue v.1.0.2 (Sollis et al, 2023). Candidate genes showed a significant (P-value = $1.4 \times 10^{-3}$) enrichment for blood cells count traits.

Finally, we investigated which TFs are disrupted by candidate variants in the enhancers of candidate genes. They showed significant enrichment for abnormal blood cell morphology/development (Bonferroni-adjusted P-value = $6.881 \times 10^{-29}$) and other related mouse phenotype terms (Table S6). Moreover, they broadly overlapped with TFs previously associated with NSCLC and different stages of chemotherapy-induced response (Vishnoi et al, 2020; Wang et al, 2021; Otálora-Otálora et al, 2023).

## Discussion

The establishment of Capture-C enabled the functional annotation of noncoding GWAS variants (Zhang & Lupski, 2015). With the development of different variations of capture Hi-C technology, most studies are still concentrated on the limited parts of the genome associated with the phenotypes of interest (Orozco et al, 2022). In this article, we performed a non-hypothesis-driven study linking SNVs associated with differing levels of chemotherapy-induced myelosuppression with their target genes at the whole-genome level. We take advantage of the HiCap technology (Sahlén et al, 2015) to prioritize candidate SNVs. Based on a 4-cutter DpnII enzyme, HiCap results in a close to single-enhancer resolution (860 bp). This precision is crucial for in vivo validations based on CRISPR genomic and epigenetic editing technologies, where correct gRNA targeting plays an important role. During our validations of detected enhancers in cells, only gRNAs targeting their centers affected the expression of targeted genes. Moreover, methods such as HiCap can identify causal variants, including those in linkage disequilibrium (Åkerborg et al, 2019); therefore, it can identify variants contributing to cellular phenotypes (Zaugg et al, 2022).

The main novelty of our approach is considering the interactome dynamics of normal versus, in this case, drug-treated cells. Whereas bulk RNA-seq profiles could not separate samples because of the stochastic RNA response upon treatments (Aissa et al, 2021), HiCap overcame this phenomenon and robustly separated samples. Our experimental design also facilitates a more comprehensive functional annotation of SNPs related to myelosuppression predisposition, whereas allowing the general description of DI genes.

Importantly, in this study, we overlap SNVs of patients with relevant enhancers, however, derived from model blood stem-like cells, limiting the discovery of private enhancers for a patient.

Taking advantage of the experimental setup, we propose a new network algorithm to analyze promoter capture Hi-C data, prioritizing DI genes based on the interactome changes between two states. We show that DI genes consistently provide more information about myelosuppression predisposition genetics than DE genes. Numerous relevant genes, such as ETV6 and MPL, appeared to be DI, but not DE at bulk RNA-seq level. Nevertheless, in some cases, DI genes supported DE genes, for example, in the described case of ERG. Our results prove that genomic loop-based assays could rescue enhancer-mediated effects on gene expression, which can be lost during transcript abundance calculations in bulk RNA-sequencing experiments because of averaging out the expression levels over cells.

Finally, we highlight the importance of DI genes by showing their enrichment for myelosuppression level-associated candidate SNPs, suggesting enhancer mutations may modulate or entirely disrupt promoter–enhancer interactions. We present a catalog of candidate genes, connected to them candidate SNVs, and TFs disrupted by these mutations, emphasizing the robustness of our approach to study promoter–enhancer dysregulation at different levels. We expect these results to make the next step toward establishing personalized medicine.

## Materials and Methods

### Carboplatin/gemcitabine treatments and cell lines

Three human cell line models were used, including two with megakaryocyte-like properties: CMK (ACC-392) (Komatsu et al, 1989; Sato et al, 1989) and MOLM-1 (ACC-720) (Matsuo et al, 1991; Ogawa et al, 1996; Drexler et al, 1999) from the Leibniz-Institute DSMZ–German Collection of Microorganisms and Cell Cultures, and one with general myelogenous properties: K-562 (CCL-243) (Lozzio & Lozzio, 1975; 1979; Lozzio et al, 1981), from the American Type Culture Collection. Cells were cultured and exposed to carboplatin or gemcitabine, as previously explained (Björn et al, 2020a). In short, duplicates of 10 million cells in 15 ml RPMI 1640 supplemented with 10% FBS were treated for 24 h with the 72-h $IC_{50}$ concentration of carboplatin, gemcitabine (both from Toronto Research Chemicals) or no drug (as a non-treated control). The drug concentrations used for K-562, CMK, and MOLM-1 were 14, 25, and 35 ng/ml for gemcitabine, and 30, 1.6, and 14 µg/ml for carboplatin, respectively. Only a minor reduction in cell viability was seen using the MTT assay at 24 h of incubation (Björn et al, 2020a). Subsequent laboratory procedures for RNAseq and HiCap (using the same cells) were immediately initiated to be as good of a snapshot of the cellular processes induced by the treatments as possible.

### RNA-sequencing, alignment, and read summarization

RNA-seq data from our previous study (Björn et al, 2020a), generated from 1 ml of cell suspension from each of the treatments,

were complemented with another round of sequencing of the previously prepared libraries using the same sequencing approach: HiSeq 2500 (Illumina) with HiSeq Rapid SBS Kit v2 chemistry and a 1 × 80 setup at Science for Life Laboratory (SciLifeLab), to increase the number of reads. Data were then merged. Briefly, TrimGalore! version 0.6.1 (Krueger, 2021) utilizing cutadapt version 2.3 (Martin, 2011) was used for quality and adapter trimming, STAR version 2.7.2b (Dobin et al, 2013) was used for alignment to GRCh37, RSEM package (Li & Dewey, 2011) version 2 was used to summarize the number of unique mapping reads per gene region (Table S1). The data quality was monitored using FastQC version 0.11.9 (Andrews, 2010), QualiMap version 2.2.1 (García-Alcalde et al, 2012), and MultiQC version 1.9 (Ewels et al, 2016).

## Gene expression and enrichment analysis

The summarized read counts were further analyzed in R version 4.0.3 (R Core Team, 2021). Reads per kilobase million, counts per million, and TPM were calculated. Analysis of differentially expressed (DE) genes was conducted using edgeR version 3.18.1 (Robinson et al, 2010; McCarthy et al, 2012) and the TMM normalization method (Robinson et al, 2010). DE genes were extracted when comparing non-treated and carboplatin or gemcitabine-treated cells for the three cell lines. KEGG pathway and Gene Ontology (GO) enrichment analyses of various sets of genes were performed using the ToppGene suite (Chen et al, 2009) and the R package clusterProfiler version 3.12.0 (Yu et al, 2012), and StringDB (Szklarczyk et al, 2021), was used for analysis.

## SNP selection

In our previous study, we analyzed 215 exomes of NSCLC patients (Björn et al, 2020b). All of them received at least one cycle of carboplatin/gemcitabine treatment, which was the standard of care for NSCLC patients at the time and place of the study. Of these, 96 patients were selected based on extreme toxicity, either low or high neutropenia, leukopenia, and thrombocytopenia. We classified patients into two groups (low and high toxicities, LT and HT, respectively) via unsupervised k-means clustering using blood cell count values of the three above phenotypes as input (Fig S10). The whole genomes of these 96 individuals were sequenced and analyzed (Björn et al, 2020a).

In this study, we overlapped SNVs detected in the 96 individuals (a total of 8,072,672 SNVs) with enhancer marks for blood cell lines collected from ENCODE (The ENCODE Project Consortium, 2012) and Fantom5 (Lizio et al, 2015) datasets available in 2017. In total, 3.27 million SNVs overlapped with at least one enhancer mark (Table S7).

## HiCap probe design

HiCapTools (Anil et al, 2018) was used to design probes against 22,171 promoters and 2,965 selected SNVs using default settings (Table S8). We selected 2,965 SNVs (the maximum number of variants we could target given the probe set size), with the largest frequency difference between patient groups and overlap with the multitude of enhancer marks to be included in the sequence probe capture set used for HiCap experiments. The probe set included

2,900 regions with no known promoter/enhancer activity. These regions were then used to generate the background interaction distribution and assign statistical significance for each observed proximity.

## HiCap library preparation

The remaining 14 ml of cell suspension from each cell sample was used for HiCap, as previously explained (Åkerborg et al, 2019; Zhigulev et al, 2022b). This method is developed by us (Sahlén et al, 2015) and others (Mifsud et al, 2015) and provides high-resolution interaction data over genomic regions by hybridizing Hi-C material to probes targeting certain regions of interest, enabling the study of individual promoter–enhancer interactions. Briefly, the method starts with cross-linking DNA–protein–DNA complexes with formaldehyde, followed by roughly cutting DNA across the genome into ~700-bp pieces using restriction endonucleases. Spatially close fragments are then ligated before capturing promoter–enhancer sequences using probes located in known genes' promoters or probes containing selected SNPs associated with ADRs. Captured libraries were sequenced using HiSeq 2500 (Illumina) with HiSeq Rapid SBS Kit v2 chemistry and a 1 × 80 setup at Science for Life Laboratory (SciLifeLab). Table S9 shows the sequencing statistics for HiCap libraries. Lastly, sequencing data are analyzed for significant interactions.

## Chromatin interaction calling

HiCapTools (Anil et al, 2018) was used to call interactions in all samples (Table S2). We required at least four pairs supporting each interaction. We then used three Bonferroni-adjusted *P*-value cutoffs (0.1, 0.01, 0.001) to filter interactions. We evaluated adjusted *P*-value cutoffs using the fold enrichment of distal regions for the publicly available enhancer datasets (Table S3), calculated by the BEDTools package version 2.30.0 (Quinlan & Hall, 2010). We used the adjusted *P*-value cutoff of 0.01 for all analyses throughout the article, except the network generation, where the threshold was 0.001. HiCap results were mapped to GRCh37.

## CRISPR validation

Several PIRs were validated using the enCRISPRi-LK system (Li et al, 2020). Briefly, gRNAs (Table S10) targeting different parts of PIRs were cloned into the Lenti-sgRNA(MS2)_MCP-KRAB-IRES-zsGreen1 (#138460) backbone plasmid with BsmBI-v2 (NEB) Golden Gate assembly, using oligos ordered from Integrated DNA Technologies. Lentiviral particles containing either tet-inducible dCas9-LSD1 (#92362), inducer TetON3G-BFP (assembled from #128061 and #120577 plasmids using BamHI-HF and MluI-HF [both from NEB]), or targeting gRNA-KRAB plasmid were separately obtained by transfection of 293T cells using Lipofectamine 3000 Transfection Reagent (Thermo Fisher Scientific) and second generation lentiviral plasmids psPAX2 (#12260) and pMD2.G (#12259) according to the manufacturer's instructions. A stable K-562-dCas9-LSD1-TetON3G-BFP cell line was obtained by transducing K-562 cells with the lentiviral particles for 6 h before changing the medium. Finally, gRNA-KRAB constructs were transduced to the previously obtained stable cell

line. dCas9 expression was induced with 1 μg/ml of doxycycline 72 h before FACS at the Biomedicum Flow cytometry Core facility (Karolinska Institutet). After sorting for BFP, zsGreen, and mCherry expressing cells, RNA was extracted with the RNeasy Plus Mini kit. RNA integrity and yield were assessed by Bioanalyzer 2100 (RNA 6000 Pico Kit; Agilent). Gene expression changes were evaluated by triplicate RT-qPCR using DreamTaq Hot Start DNA Polymerase (Thermo Fisher Scientific) and an EvaGreen dye (Biotium).

### Interactome network generation and calculation of network parameters

To generate interactome networks, we considered each interacting promoter or enhancer as a node in the network. We then connected two nodes if they had a significant interaction in at least one replicate. We used the *igraph* package version 1.4.2 (Csardi & Nepusz, 2006) distributed under R to generate the networks. We then created a summary of interactions for each node or gene by summing its interactions. We generated nine networks in total (Table S5).

To calculate the Jaccard index (JI) and overlap coefficient (OCE) of a node between the two networks (A, B), we used the following formulas:

$$JI_x = \frac{A \cap B}{A \cup B}$$

$$OCE_x = \frac{A \cap B}{\min(A, B)}$$

where A and B denote the number of neighbors of node x in networks A and B, respectively.

The network is generated using non-replicated interactions (i.e., those that are called interactions in one of the replicates); therefore, JI and OCE parameters reflect the amount of replicated novel (gained) or lost interactions. Note that a node with a JI value less than one can have an OCE value equal to one. This is because OCE requires that the number of common nodes is more than the number of nodes in the smaller network. For example, a node with two connections in the untreated condition could lose one of its connections upon treatment and not gain any additional connection. The JI would be 0.5, but OCE would be 1 because the size of the network with minimum number of nodes is equal to the number of common nodes. Therefore, OCE is a stricter measure of connectivity change than JI.

We then defined differentially interacting (DI) genes as those OCE < 1 and JI <= 0.5. We excluded cases with JI = 0 to eliminate genes gaining or losing only one interaction. We only took genes with TPM > 0.2.

### Allele count difference calculations

We calculated the allele count (AC) and frequencies (AF) of each alternative allele for both HT and LT groups of patients. We then took the absolute difference between LT and HT groups for AC and AF:

$$AFdiff = \frac{abs(AF.LT - AF.HT)}{AF.LT + AF.HT}$$

$$ACdiff = \frac{abs(AC.LT - AC.HT)}{AC.LT + AC.HT}$$

For candidate variants, we required them to be present in at least two individuals at a 25% AC and AF difference between the two groups. In short, that is represented by the following formula:

$$(AFdiff > ACEthr, ACdiff > ACEthr, abs(AC.LT - AC.HT) > 1)$$

where allele counts difference threshold, ACEthr = 0.25.

The variants not fulfilling the following criteria are deemed as "rest".

### TF affinity predictions

To assess the potential effect of SNVs on the TF-binding affinity, we used the motifbreakR package (Coetzee et al, 2015). It directly weights the score by the importance of the position within a particular motif in a tested collection. We used a motifbreakR_motif object derived from MotifDb package version 1.40.0 (Shannon & Richards, 2023)—the collection of 2,817 position frequency matrices from four sources: ENCODE-motif, FactorBook, HOCOMOCO, and HOMER.

## Data Availability

Whole-genome sequencing datasets analyzed during the current study are not publicly available because of reasons of sensitivity but are available from the corresponding author upon reasonable request. Raw sequence reads of RNA-seq and HiCap are submitted to NCBI Sequence Read Archive (PRJNA1012445).

### Ethics statement

The 96 whole-genome sequences of NSCLC patients were diagnosed between 2006 and 2008 at Karolinska University Hospital, Stockholm, Sweden, and included after providing written informed consent in accordance with the Helsinki Declaration. The study received ethical approval from the Regional Ethics Committee in Stockholm (DNR-03-413 with amendment 2016/258-32/1). These patients are part of the material included in previously published studies (Björn et al, 2020a, 2020b; Svedberg et al, 2020).

## Supplementary Information

# Acknowledgements

We are grateful to Joakim Lundeberg (Royal Institute of Technology—KTH, Stockholm, Sweden) for fruitful discussions. We are also thankful to Bernhard Schmierer (SciLifeLab CRISPR Functional Genomics Unit, Solna, Sweden) for providing the 293T cell line and Ivan Kulakovskiy (Institute of Protein Research, Russian Academy of Sciences, Pushchino, Russia) for his suggestions on analyzing TF-binding affinity. The authors acknowledge support from the National Genomics Infrastructure in Stockholm funded by Science for Life Laboratory, the Knut and Alice Wallenberg Foundation, and the Swedish Research Council, and SNIC/Uppsala Multidisciplinary Center for Advanced Computational Science for assistance with massively parallel sequencing and access to the UPPMAX computational infrastructure. They also acknowledge the Biomedicum Flow Cytometry core facility (Karolinska Institutet), supported by KI/SLL, for providing cell sorting services and the KIGene core facility (Karolinska Institutet) for providing sequencing services. This project has received funding from the European Union's Horizon 2020 Research and Innovation Programme under the Marie Sklodowska-Curie Grant Agreement No 860002. The study was also funded by the Swedish Cancer Society, The Swedish Childhood Cancer Fond (barncancerfonden), and ALF Grants, Region Östergötland, and by the Swedish Research Council (Grant Agreement No. 78081). The information contained in this presentation reflects only the authors' views. REA and EC are not responsible for any use that may be made of this information. This article is dedicated to the memory of EM, who passed away while the aricle was in preparation.

## Author Contributions

A Zhigulev: data curation, software, formal analysis, visualization, project administration, and writing—original draft, review, and editing.
Z Norberg: validation and writing—review and editing.
J Cordier: validation and writing—review and editing.
R Spalinskas: investigation and writing—review and editing.
H Bassereh: software, formal analysis, and writing—review and editing.
N Björn: formal analysis, investigation, and writing—review and editing.
S Pradhananga: data curation and writing—review and editing.
H Gréen: conceptualization, resources, supervision, funding acquisition, methodology, and writing—review and editing.
P Sahlén: conceptualization, resources, data curation, supervision, funding acquisition, visualization, methodology, project administration, and writing—original draft, review, and editing.

## Conflict of Interest Statement

P Sahlén holds a HiCap patent (EP2984182A1).

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
