## [Reviewer comments · Life Science Alliance]

Enhancer mutations modulate the severity of chemotherapy-induced myelosuppression

Artemy Zhigulev, Zandra Norberg, Julie Cordier, Rapolas Spalinskas, Hassan Bassereh, Niclas Björn, Sailendra Pradhananga, Henrik Gréen and Pelin Sahlén

DOI: <https://doi.org/10.26508/lsa.202302244>

Corresponding author(s): *Pelin Sahlén (Royal Institute of Technology)*

Review Timeline:

Submission Date:	2023-06-30
Editorial Decision:	2023-09-07
Revision Received:	2023-11-15
Editorial Decision:	2023-12-11
Revision Received:	2023-12-19
Accepted:	2023-12-21

Scientific Editor: *Eric Sawey, PhD*

Transaction Report:

September 7, 2023

Re: Life Science Alliance manuscript #LSA-2023-02244-T

Dr Pelin Sahlén
KTH - Royal Institute of Technology
Science for Life Laboratory Stockholm, School of Biotechnology
Tomtebodavagen 23A
Solna 17165
Sweden

Dear Dr. Sahlén,

Thank you for submitting your manuscript entitled "Enhancer mutations modulate the severity of chemotherapy-induced myelosuppression" to Life Science Alliance. The manuscript was assessed by expert reviewers, whose comments are appended to this letter. We invite you to submit a revised manuscript addressing the Reviewer comments.

Thank you for this interesting contribution to Life Science Alliance. We are looking forward to receiving your revised manuscript.

Sincerely,

B. MANUSCRIPT ORGANIZATION AND FORMATTING:

Reviewer #1 (Comments to the Authors (Required)):

In this study, the authors aim to predict the importance of distal (well-characterized and specific) cis-regulatory regions for myelosuppression during chemotherapy treatment with carboplatin and gemcitabine in NSCLC. To achieve this, they perform Capture HiC in three AML cell lines that exhibit features similar to hematopoietic stem cells, both with and without treatment with carboplatin and gemcitabine. They subsequently characterize differential gene expression and, importantly, differential promoter interactions with distal cis-regulatory regions.

From a technical standpoint, the study is well-executed. The methods are explained in a manner that enables readers to comprehend them. However, there are certain aspects that require further clarification:

1. It is reasonable to expect that AML cell lines would display DI dynamics following 24 hours of treatment with toxic chemotherapies. Also, it's essential to demonstrate that the cellular morphology remains unaffected, ensuring there is no selective bias induced by the treatment. How was the dosage chosen (i.e., one-third of the 72-hour IC50 dose)? The authors should demonstrate, whether cell morphology remained unaltered at the 24-hour treatment mark, coinciding with the Capture HiC procedure. This can be achieved through viability assessments (such as Annexin-V staining and cell cycle analysis), proliferation evaluations (including cell count or MTT assays), and differentiation assays (employing Flow Cytometry for differentiation markers). Additionally, it would be insightful for the readers to understand why the experiments were not conducted using a hematopoietic cell line or primary hematopoietic stem cells.
2. What is the extent of overlap between the SNVs matching differentially interacting cis-regulatory regions and those that were overrepresented in "hematopoietic" and/or "myeloid" gene sets? Do these sets share similarities, or are they distinct? It's worth noting that the supplementary tables are missing from the submitted manuscript, which would have provided valuable information, particularly Supplementary Table 10.
3. From a clinical perspective, Cisplatin tends to be favored over Carboplatin in various cancer types due to its superior efficacy. However, there are exceptions, such as ovarian and aggressive neuroendocrine cancers, where Carboplatin is preferred. I suggest the authors slightly modify their introduction to accurately reflect the use of Platin doublets (either Cis- or Carboplatin) along with 3rd generation chemotherapeutics, like Gemcitabine, as a standard/common "backbone" treatment across a broad spectrum of cancers. It's important to acknowledge that Carboplatin has a higher bone marrow toxicity compared to other types of Platin.

Reviewer #2 (Comments to the Authors (Required)):

In this manuscript by Zhigulev and colleagues, the authors investigate the possible role of enhancer 'mutations' in terms of susceptibility to chemotherapy induced myelosuppression. They use HiCap to investigate promoter interactions in 3 cell lines treated with classical NSCLC chemotherapy drugs carboplatin and gemcitabine. The concept of altered enhancer usage under selective pressure of therapy is not entirely new, but has generally been conducted in locus specific ways rather than genome wide as performed here. The idea that enhancer mutations might alter this response is interesting, but the functional significance of such 'mutations' is not entirely fleshed out here and leaves the findings as more observational/preliminary. The methodological approach is probably more interesting to the scientific community than the findings themselves.

The title uses the term 'mutations' which immediately insinuates these SNVs as somatic, yet most of the SNVs in their previous patient study are germline SNPs. Can the authors please clarify which of the enhancer variants in the cell lines are likely somatic (ie not identified in germline variant databases)? Given the bone marrow compartment in patients with lung cancer is not diseased, how relevant are these 'mutations'?

Can the authors clarify the advantage of HiCAP over other promoter capture methods such as Capture-C? The 'targeted variants HiCap' probe design is very vague. How were the 2,965 selected SNVs selected? Are the probes able to efficiently capture only the variant allele over the wild type allele if there is a single SNV?

How representative are the promoter-enhancer interactions at steady state in the 3 cell lines vs normal hematopoietic progenitor cells? ie are these transformed leukemic cells really a good model for studying drug toxicity of normal hematopoietic cells? What was the timing between drug treatments and RNAseq/HiCap? What are the drug IC50s to the 3 cell lines and what doses were used for the experiments? It is important to ascertain what percentage of cells were undergoing apoptosis at the time-points used for these experiments.

One of my major concerns is Fig S3: the PCA shows a very wide separation of the 2 replicates, in some cases more so than treated vs untreated. Can the authors please comment on this. If the authors compare DEGs between replicate 1 vs 2 in untreated conditions, how many genes are significantly different? In the same vein, what about significantly different promoter-enhancer interactions between the two replicates? This wide variation in data is worrying when it comes to study reproducibility.

The OCE/JI index is not able to distinguish genes that gained all new interactions vs those that lost all interactions. Would it not be interesting to differentiate genes according to these outcomes? Are there any interesting features of enhancers that are lost vs those that are gained?

Alterations in enhancer usage is likely to be dictated by TF occupancy at those sites. If the authors look at motif calling (eg using MEME) at PIRs of DI genes that are newly activated on chemotherapy treatment, they may be able to identify master TFs that dictate these interactions. For example, one would hypothesise TP53 is activated under chemotherapy and occupies new sites creating new enhancer promoter interactions (although these cell lines may be TP53 mutated.)

The 233 DI genes identified by the authors are not followed up in any way. Are there any available data, for instance from GWAS, that might support that these genes and their enhancers are in any way relevant to chemotherapy induced myelosuppression?

Minor point

Term 'in vivo' in abstract means is often interpreted as in a living animal to most biologists.

Can we see the promoter enhancer loops for the genes tested in fig 1 B/C

'NFE2 enhances the expression of the hematopoietic master regulators SCL/TAL1 and GATA2.' This statement is not really relevant to the paper as used only as a negative control.

Enhancer mutations modulate the severity of chemotherapy-induced myelosuppression

Reviewers' Comments

Please find attached the revised manuscript ID #LSA-2023-02244-T entitled “Enhancer mutations modulate the severity of chemotherapy-induced myelosuppression” which we now resubmit to your journal, Life Science Alliance. We thank the reviewers and the editors for taking the time to read and give comments and suggestions on our initial version of this manuscript. The reviewers' comments are included below, and our response/changes are indicated.

Reviewer #1 (Comments to the Authors (Required)):

In this study, the authors aim to predict the importance of distal (well-characterized and specific) cis-regulatory regions for myelosuppression during chemotherapy treatment with carboplatin and gemcitabine in NCSLC. To achieve this, they perform Capture HiC in three AML cell lines that exhibit features similar to hematopoietic stem cells, both with and without treatment with carboplatin and gemcitabine. They subsequently characterize differential gene expression and, importantly, differential promoter interactions with distal cis-regulatory regions.

From a technical standpoint, the study is well-executed. The methods are explained in a manner that enables readers to comprehend them. However, there are certain aspects that require further clarification:

Q1: It is reasonable to expect that AML cell lines would display DI dynamics following 24 hours of treatment with toxic chemotherapies. Also, it's essential to demonstrate that the cellular morphology remains unaffected, ensuring there is no selective bias induced by the treatment. How was the dosage chosen (i.e., one-third of the 72-hour IC50 dose)? The authors should demonstrate, whether cell morphology remained unaltered at the 24-hour treatment mark, coinciding with the Capture HiC procedure. This can be achieved through viability assessments (such as Annexin-V staining and cell cycle analysis), proliferation evaluations (including cell count or MTT assays), and differentiation assays (employing Flow Cytometry for differentiation markers). Additionally, it would be insightful for the readers to understand why the experiments were not conducted using a hematopoietic cell line or primary hematopoietic stem cells.

Response: We are very pleased that our reviewer asked regarding this part of the design in our study. The dosage corresponded to the 72-hour IC50 concentration (for K-562, CMK, and MOLM-1 cells, concentrations were 14 ng/ml, 25 ng/ml, and 35 ng/ml for gemcitabine, and 30 µg/ml, 1.6 µg/ml, and 14 µg/ml for carboplatin, respectively). As our reviewer points out, it is crucial not to induce large cellular morphological changes when exposing the cells to the drug

concentration selected. We, therefore, also performed dose-response curves for the drugs with a 24h MTT-assay (Björn et al, 2020a). The results showed that we had only a minor reduction in cell viability at 24h (10-15%). We now added this information to the Materials and Methods section, “Carboplatin/gemcitabine treatments and cell lines” paragraph.

AML cell lines are a good and convenient surrogate model that is readily available in many research laboratories (Skopek et al, 2023). The cell lines can be stimulated to differentiate towards other more specific hematopoietic cells. Patient primary cells are harder to come by and more challenging to handle in the lab than these surrogate lineages. However, we could hint that we have previous single-cell RNA-seq publication using hematopoietic stem and progenitor cells treated with these drugs (Björn et al, 2020b). But at that moment in time, we did not perform HiCap. The Results section, “HiCap depicts interactome dynamics of blood stem-like cells upon drug exposure” paragraph was accordingly adjusted.

Q2: What is the extent of overlap between the SNVs matching differentially interacting cis-regulatory regions and those that were overrepresented in “hematopoietic” and/or “myeloid” gene sets? Do these sets share similarities, or are they distinct? It’s worth noting that the supplementary tables are missing from the submitted manuscript, which would have provided valuable information, particularly Supplementary Table 10.

Response: We thank our reviewer for this suggestion. We further investigated the set of candidate genes in our manuscript. They showed a significant (p -value = $1.4E-3$) enrichment for genes associated with blood cells (lymphocytes, erythrocytes, platelets, and neutrophils) count traits derived from the NHGRI-EBI GWAS Catalogue (Sollis et al, 2023). We now added these results to the “Genes connected to candidate enhancer variants tend to differentially interact upon drug exposure” paragraph.

Q3: From a clinical perspective, Cisplatin tends to be favored over Carboplatin in various cancer types due to its superior efficacy. However, there are exceptions, such as ovarian and aggressive neuroendocrine cancers, where Carboplatin is preferred. I suggest the authors slightly modify their introduction to accurately reflect the use of Platin doublets (either Cis- or Carboplatin) along with 3rd generation chemotherapeutics, like Gemcitabine, as a standard/common “backbone” treatment across a broad spectrum of cancers. It’s important to acknowledge that Carboplatin has a higher bone marrow toxicity compared to other types of Platin.

Response: As our reviewer suggests, this is an excellent point to cover in the introduction. Carboplatin is associated with a higher risk of myelosuppression and neurotoxicity compared to cisplatin, which is associated with a higher rate of nausea, vomiting, nephrotoxicity, and ototoxicity. The 3rd generation chemotherapy is also used as a backbone of classical chemotherapies. The Background section was appropriately modified.

Reviewer #2 (Comments to the Authors (Required)):

In this manuscript by Zhigulev and colleagues, the authors investigate the possible role of enhancer ‘mutations’ in terms of susceptibility to chemotherapy induced myelosuppression. They use HiCap to investigate promoter interactions in 3 cell lines treated with classical NSCLC chemotherapy drugs carboplatin and gemcitabine. The concept of altered enhancer usage under selective pressure of therapy is not entirely new, but has generally been conducted in locus specific ways rather than genome wide as performed here. The idea that enhancer mutations might alter this response is interesting, but the functional significance of such ‘mutations’ is not entirely fleshed out here and leaves the findings as more observational/preliminary. The methodological approach is probably more interesting to the scientific community than the findings themselves.

Q1: The title uses the term ‘mutations’ which immediately insinuates these SNVs as somatic, yet most of the SNVs in their previous patient study are germline SNPs. Can the authors please clarify which of the enhancer variants in the cell lines are likely somatic (ie not identified in germline variant databases)? Given the bone marrow compartment in patients with lung cancer is not diseased, how relevant are these ‘mutations’?

Response: We thank our reviewer for pointing out this critical difference. We only focused on germline mutations in this study and explained this now clearly in the Introduction section.

Q2: Can the authors clarify the advantage of HiCAP over other promoter capture methods such as Capture-C? The ‘targeted variants HiCap’ probe design is very vague. How were the 2,965 selected SNVs selected? Are the probes able to efficiently capture only the variant allele over the wild type allele if there is a single SNV?

Response: Promoter capture techniques can be based either on 3C or Hi-C. Capture-C is a 3C-based method. Capture-C libraries are dominated by uninformative (un-ligated) fragments due to the lack of biotin selection step of ligated products, resulting in fewer informative reads compared to Hi-C-based methods (>95% vs 26–46% correspondingly) (Sahlén et al, 2015). HiCap is based on Hi-C and, therefore, includes a minimal amount of unligated fragments. Moreover, promoter capture Hi-C approaches usually differ by resolutions. In PCHi-C method the genome is fragmented using HindIII, yielding a resolution of 3.4 kb (Mifsud et al, 2015; Schoenfelder et al, 2015). However, HiCap is based on a 4-cutter restriction enzyme (DpnII), resulting in a close to single enhancer resolution (860 bp). We now clarified the advantages of HiCap in the Introduction section.

The 2,965 SNVs were selected based on the largest allelic frequency difference between low and high patient toxicity groups and overlap with the multitude of enhancer marks (Materials and

Methods, “HiCap probe design” paragraph). The probe set size limited the number of included SNVs; therefore, we could only take the top SNPs based on the space left in the design after including all the promoters and negative controls.

In this study, we did not aim at phasing HiCap data, as we combined variants from real patients with HiCap on model cell lines. That is the exact reason why we designed probes to flank the broader area around selected SNVs. However, in our ongoing stage II study of NCSLC, WGS and HiCap are performed on the same patients, and probes are designed to overlap SNVs. The probes are still too long and will not be able to capture haplotypes, but we can phase the data based on the HiCap sequence itself to a limited extent.

Q3: How representative are the promoter-enhancer interactions at steady state in the 3 cell lines vs normal hematopoietic progenitor cells? ie are these transformed leukemic cells really a good model for studying drug toxicity of normal hematopoietic cells?

What was the timing between drug treatments and RNAseq/HiCap?

What are the drug IC50s to the 3 cell lines and what doses were used for the experiments? It is important to ascertain what percentage of cells were undergoing apoptosis at the time-points used for these experiments.

Response: We are very pleased that our reviewer asked regarding this part of the design in our study. Transformed leukemic cells are a good and convenient surrogate model readily available in many research laboratories (Skopek et al, 2023). The cell lines can be stimulated to differentiate towards other more specific hematopoietic cells. Patient primary cells are harder to come by and more challenging to handle in the lab than these surrogate lineages. However, we could hint that we have previous single-cell RNA-seq publication using hematopoietic stem and progenitor cells treated with these drugs (Björn et al, 2020b). But at that moment in time, we did not perform HiCap. The Results section, “HiCap depicts interactome dynamics of blood stem-like cells upon drug exposure” paragraph was accordingly adjusted.

Directly after the 24-hour incubation with carboplatin/gemcitabine, the cells were harvested, and the subsequent laboratory procedures for RNAseq and HiCap (using the same cells) were immediately initiated to be as good of a snapshot of the cellular processes induced by the treatments as possible. The Materials and Methods section, “Carboplatin/gemcitabine treatments and cell lines” paragraph was accordingly adjusted.

The dosage corresponded to the 72-hour IC50 concentration (for K-562, CMK, and MOLM-1 cells, concentrations were 14 ng/ml, 25 ng/ml, and 35 ng/ml for gemcitabine, and 30 µg/ml, 1.6 µg/ml, and 14 µg/ml for carboplatin, respectively). As our reviewer points out, it is crucial not to induce large cellular morphological changes when exposing the cells to the drug concentration selected. We, therefore, also performed dose-response curves for the drugs with a 24h MTT-assay (Björn et al, 2020a). The results showed that we had only a minor reduction in cell viability at 24h (10-15%). We now added this information to the Materials and Methods section, “Carboplatin/gemcitabine treatments and cell lines” paragraph.

Q4: One of my major concerns is Fig S3: the PCA shows a very wide separation of the 2 replicates, in some cases more so than treated vs untreated. Can the authors please comment on this. If the authors compare DEGs between replicate 1 vs 2 in untreated conditions, how many genes are significantly different? In the same vein, what about significantly different promoter-enhancer interactions between the two replicates? This wide variation in data is worrying when it comes to study reproducibility.

Response: Firstly, we thank our reviewer for raising this point. This is the reason why we also included Fig S2, which shows an excellent correlation between the replicates, and we did not find any DE genes between the replicates of the untreated conditions for all cell types. Upon induction, stochastic RNA response could cause such patterns in the expression data (Aissa et al, 2021). However, HiCap overcame this phenomenon and robustly separated samples.

Q5: The OCE/JI index is not able to distinguish genes that gained all new interactions vs those that lost all interactions. Would it not be interesting to differentiate genes according to these outcomes? Are there any interesting features of enhancers that are lost vs those that are gained?

Response: These extreme events are indeed exciting cases of interactome changes. We further looked into the cases when candidate genes gain or lose all the interactions upon drug induction. We generated separate networks for each replicate and calculated network parameters accordingly. This increased the robustness of network parameters by excluding unreplicated interactions to be counted as changes (for more details, please see the Materials and Methods section, “Interactome network generation and calculation of network parameters” paragraph). Therefore, considering only replicated interactions, no candidate genes gain all interactions, and eight candidate genes lose all interactions upon the treatment. Therefore, most candidate genes have varying levels of interaction rewiring upon treatment. Due to the low number of such extreme cases, it is impossible to draw statistically significant conclusions on the lost/gained enhancer sequences. However, we believe that WGS and HiCap performed on the same patients in our phase II study will give us enough power for this type of analysis.

Q6: Alterations in enhancer usage is likely to be dictated by TF occupancy at those sites. If the authors look at motif calling (eg using MEME) at PIRs of DI genes that are newly activated on chemotherapy treatment, they may be able to identify master TFs that dictate these interactions. For example, one would hypothesise TP53 is activated under chemotherapy and occupies new sites creating new enhancer promoter interactions (although these cell lines may be TP53 mutated.)

Response: We thank our reviewer for this excellent suggestion. The motifbreakR package predicts the changes in TF binding affinity between alleles. We investigated which TF binding

sites are disrupted by candidate variants in regulatory elements of candidate genes. This set of TFs was enriched for abnormal blood cell morphology/development (Bonferroni adjusted p-value = 6.881E-29) and other related Mouse Phenotype terms (Table S6). Additionally, they broadly overlapped with TFs previously associated with NSCLC and different stages of chemotherapy-induced response (Vishnoi et al, 2020; Wang et al, 2021; Otálora-Otálora et al, 2023). We now added these results to the “Genes connected to candidate enhancer variants tend to differentially interact upon drug exposure” paragraph.

Q7: The 233 DI genes identified by the authors are not followed up in any way. Are there any available data, for instance from GWAS, that might support that these genes and their enhancers are in any way relevant to chemotherapy induced myelosuppression?

Response: We thank our reviewer for the suggestion. We further investigated the set of candidate genes accordingly. They showed a significant (p-value = 1.4E-3) enrichment for blood cells (lymphocytes, erythrocytes, platelets, and neutrophils) count traits derived from the NHGRI-EBI GWAS Catalogue (Sollis et al, 2023). We now added these results to the “Genes connected to candidate enhancer variants tend to differentially interact upon drug exposure” paragraph.

Minor point

Term ‘in vivo’ in abstract means is often interpreted as in a living animal to most biologists. “In vivo” was replaced by “in living cells”.

Can we see the promoter enhancer loops for the genes tested in fig 1 B/C

Figure S1 with promoter-enhancer loops for *TGFBR1* and *ANAPC4* was added.

‘NFE2 enhances the expression of the hematopoietic master regulators SCL/TAL1 and GATA2.’

This statement is not really relevant to the paper as used only as a negative control.

The sentence is deleted.

References

Aissa AF, Islam ABMMK, Ariss MM, Go CC, Rader AE, Conrardy RD, Gajda AM, Rubio-Perez C, Valyi-Nagy K, Pasquinelli M *et al* (2021) Single-cell transcriptional changes associated with drug tolerance and response to combination therapies in cancer. *Nat Commun* 12: 1628.

doi:10.1038/s41467-021-21884-z.

Björn N, Badam TV, Spalinskas R, Brandén E, Koyi H, Lewensohn R, De Petris L, Lubovac-Pilav Z, Sahlén P, Lundeberg J *et al* (2020a) Whole-genome sequencing and gene network modules predict gemcitabine/carboplatin-induced myelosuppression in non-small cell lung cancer patients. *NPJ Syst Biol Appl* 6: 25. doi:10.1038/s41540-020-00146-6.

Björn N, Jakobsen I, Lotfi K, Gréen H (2020b) Single-Cell RNA Sequencing of Hematopoietic Stem and Progenitor Cells Treated with Gemcitabine and Carboplatin. *Genes (Basel)* 11: 549. doi:10.3390/genes11050549.

Mifsud B, Tavares-Cadete F, Young AN, Sugar R, Schoenfelder S, Ferreira L, Wingett SW, Andrews S, Grey W, Ewels PA *et al* (2015) Mapping long-range promoter contacts in human cells with high-resolution capture Hi-C. *Nat Genet* 47. doi:10.1038/ng.3286.

Otálora-Otálora BA, López-Kleine L, Rojas A (2023) Lung Cancer Gene Regulatory Network of Transcription Factors Related to the Hallmarks of Cancer. *Curr Issues Mol Biol* 45: 434–464. doi:10.3390/cimb45010029.

Sahlén P, Abdullayev I, Ramsköld D, Matskova L, Rilakovic N, Lötstedt B, Albert TJ, Lundberg J, Sandberg R (2015) Genome-wide mapping of promoter-anchored interactions with close to single-enhancer resolution. *Genome Biol* 16. doi:10.1186/s13059-015-0727-9.

Schoenfelder S, Furlan-Magaril M, Mifsud B, Tavares-Cadete F, Sugar R, Javierre B-M, Nagano T, Katsman Y, Sakthidevi M, Wingett SW *et al* (2015) The pluripotent regulatory circuitry connecting promoters to their long-range interacting elements. *Genome Res* 25: 582–597. doi:10.1101/gr.185272.114.

Skopek R, Palusińska M, Kaczor-Keller K, Pingwara R, Papierniak-Wyglądała A, Schenk T, Lewicki S, Zelent A, Szymański Ł (2023) Choosing the Right Cell Line for Acute Myeloid Leukemia (AML) Research. *Int J Mol Sci* 24: 5377. doi:10.3390/ijms24065377.

Sollis E, Mosaku A, Abid A, Buniello A, Cerezo M, Gil L, Groza T, Güneş O, Hall P, Hayhurst J *et al* (2023) The NHGRI-EBI GWAS Catalog: knowledgebase and deposition resource. *Nucleic Acids Res* 51: D977–D985. doi:10.1093/nar/gkac1010.

Vishnoi K, Viswakarma N, Rana A, Rana B (2020) Transcription Factors in Cancer Development and Therapy. *Cancers (Basel)* 12: 2296. doi:10.3390/cancers12082296.

Wang L, Peng Q, Yin N, Xie Y, Xu J, Chen A, Yi J, Tang J, Xiang J (2021) Chromatin accessibility regulates chemotherapy-induced dormancy and reactivation. *Mol Ther Nucleic Acids* 26: 269–279. doi:10.1016/j.omtn.2021.07.019.

December 11, 2023

RE: Life Science Alliance Manuscript #LSA-2023-02244-TR

Prof. Pelin Sahlén
Royal Institute of Technology
Science for Life Laboratory Stockholm, School of Biotechnology
Tomtebodavagen 23A
Solna 17165
Sweden

Dear Dr. Sahlén,

Thank you for submitting your revised manuscript entitled "Enhancer mutations modulate the severity of chemotherapy-induced myelosuppression". We would be happy to publish your paper in Life Science Alliance pending final revisions necessary to meet our formatting guidelines.

- please consult our manuscript preparation guidelines <https://www.life-science-alliance.org/manuscript-prep> and make sure your manuscript sections are in the correct order
- please add ORCID ID for corresponding (and secondary corresponding) author--you should have received instructions on how to do so
- Please upload all figure files as individual ones (including Figure number), including the supplementary figure files; all figure legends should only appear in the main manuscript file
- please add your main, supplementary figure, and table legends to the main manuscript text after the references section
- please add a callout for all Figures and Supplementary Figures to your main manuscript text - Fig S1 a,b,c,d,e,f; Fig S3 a,b; Fig S7 a,b
- please rename the datasets as supplementary tables - both in their titles and in their callouts in the manuscript text

A. FINAL FILES:

B. MANUSCRIPT ORGANIZATION AND FORMATTING:

Sincerely,

Reviewer #1 (Comments to the Authors (Required)):

With the provided revision, the authors answered all reviewer questions/points. The manuscript appears consolidated, and I recommend its publication in the current form.

Reviewer #2 (Comments to the Authors (Required)):

The authors have addressed my major concerns.

December 21, 2023

RE: Life Science Alliance Manuscript #LSA-2023-02244-TRR

Prof. Pelin Sahlén
Royal Institute of Technology
Science for Life Laboratory Stockholm, School of Biotechnology
Tomtebodavagen 23A
Solna 17165
Sweden

Dear Dr. Sahlén,

Thank you for submitting your Research Article entitled "Enhancer mutations modulate the severity of chemotherapy-induced myelosuppression". It is a pleasure to let you know that your manuscript is now accepted for publication in Life Science Alliance. Congratulations on this interesting work.

DISTRIBUTION OF MATERIALS:

Again, congratulations on a very nice paper. I hope you found the review process to be constructive and are pleased with how the manuscript was handled editorially. We look forward to future exciting submissions from your lab.

Sincerely,
